# Differentiation alters stem cell nuclear architecture, mechanics, and mechano-sensitivity

Su-Jin Heo[1,2], Tristan P Driscoll[1,2], Stephen D Thorpe[3], Nandan L Nerurkar[4], Brendon M Baker[5,6], Michael T Yang[5], Christopher S Chen[5,6], David A Lee[3], Robert L Mauck[1,2]*

[1]McKay Orthopaedic Research Laboratory, Department of Orthopaedic Surgery, Perelman School of Medicine, University of Pennsylvania, Philadelphia, United States; [2]Department of Bioengineering, School of Engineering and Applied Science, University of Pennsylvania, Pennsylvania, United States; [3]Institute of Bioengineering, School of Engineering and Materials Science, Queen Mary University of London, London, United Kingdom; [4]Department of Genetics, Harvard Medical School, Harvard University, Boston, United States; [5]Department of Biomedical Engineering, College of Engineering, Boston University, Boston, United States; [6]Wyss Institute for Biologically Inspired Engineering, Harvard University, Boston, United States

*For correspondence: lemauck@mail.med.upenn.edu

Competing interests: The authors declare that no competing interests exist.

**Abstract** Mesenchymal stem cell (MSC) differentiation is mediated by soluble and physical cues. In this study, we investigated differentiation-induced transformations in MSC cellular and nuclear biophysical properties and queried their role in mechanosensation. Our data show that nuclei in differentiated bovine and human MSCs stiffen and become resistant to deformation. This attenuated nuclear deformation was governed by restructuring of Lamin A/C and increased heterochromatin content. This change in nuclear stiffness sensitized MSCs to mechanical-loading-induced calcium signaling and differentiated marker expression. This sensitization was reversed when the 'stiff' differentiated nucleus was softened and was enhanced when the 'soft' undifferentiated nucleus was stiffened through pharmacologic treatment. Interestingly, dynamic loading of undifferentiated MSCs, in the absence of soluble differentiation factors, stiffened and condensed the nucleus, and increased mechanosensitivity more rapidly than soluble factors. These data suggest that the nucleus acts as a mechanostat to modulate cellular mechanosensation during differentiation.

## Introduction

Mesenchymal stem cells (MSCs) are used in a variety of regenerative applications (*Bianco et al., 2013*). While considerable work has shown the importance of soluble differentiation factors in MSC lineage specification, recent studies have also highlighted that physical signals from the microenvironment, including substrate stiffness (*Engler et al., 2006*), cell shape (*McBeath et al., 2004*), and dynamic mechanical cues (*Huang et al., 2010a*) can influence fate decisions. However, the manner in which soluble and physical cues are integrated to inform lineage specification and commitment is only just beginning to be understood (*Guilak et al., 2009*). One potentially confounding feature is that the physical properties of MSCs themselves likely change coincident with lineage specification, and such changes might alter cellular perception of super-imposed mechanical perturbations that arise from the microenvironment.

Strain transfer to (and deformation of) the nucleus has been proposed as a direct link between mechanical inputs from the microenvironment and gene regulation (*Wang et al., 2009*). The cytoskeleton forms a mechanically continuous network within the cell and transmits extracellular mechanical signals from sites of matrix adhesion to the nucleus through specialized proteins that comprise the linker of nucleus and cytoskeleton (LINC) complex (*Haque et al., 2006*). These connections allow for direct transfer of mechanical signals to the chromatin (*Wang et al., 2009*; *Martins et al., 2012*) annscription upregulation viad can regulate intracellular signaling (*Driscoll et al., 2015*). Chromatin remodeling induced by mechanical signals depends in part on a pre-tensed (contractile) actin cytoskeleton (*Hu et al., 2005*; *Heo et al., 2016*) and can regulate gene expression (*Wang et al., 2009*; *Tajik et al., 2016*; *Shivashankar, 2011*). Together, these findings demonstrate that changes in cytoskeletal organization, connectedness to the nuclear envelope, and pre-tension in the actomyosin network all impact how cells sense and respond to mechanical signals.

Since the nucleus is the stiffest of organelles, changes in nuclear architecture might also impact how forces are transmitted through the cell. It is well established that chromatin condensation increases nuclear stiffness (*Dahl et al., 2005*), as do changes in the amount and distribution of other intra-nuclear filamentous proteins, including the lamin protein family (*Ho and Lammerding, 2012*). For example, nuclear lamins stabilize and stiffen the nuclear envelope and are regulated both by differentiation (*Lammerding et al., 2006*) and the micro-elasticity of the surrounding tissue (*Swift et al., 2013*). Mouse embryonic fibroblasts lacking lamin A/C (LMAC) have aberrant nuclear morphologies and exaggerated nuclear deformation in response to deformation of the cell (*Lammerding et al., 2004*). Knockdown of LMAC in the nuclei of differentiated cells decreases nuclear stiffness (*Pajerowski et al., 2007*), while overexpression in neutrophils decreases their ability to pass through micron-sized openings (*Davidson et al., 2014*). In addition, lamins may contribute to chromatin remodeling, gene silencing, and transcriptional activation (*Andrés and González, 2009*; *Mewborn et al., 2010*) via the action of lamin binding proteins (*Wilson and Foisner, 2010*) and their sequestration of chromatin to the nuclear periphery (*Gurudatta et al., 2010*).

As progenitor cells differentiate, a host of physical changes occur within the cell, depending on cell type and the lineage to which they are being driven. These biophysical changes extend to the nucleus, where for instance ES cell differentiation is accompanied by an increase in chromatin condensation (*Bártová et al., 2008*) leading to an increase in nuclear stiffness (*Pajerowski et al., 2007*). Lamins change during differentiation as well; mouse ES cells start expressing high levels of A-type lamin during cell differentiation, suggestive of a role in the maintenance of differentiated state. Further, chromatin reorganization mediated by lamins can enhance heterochromatin formation in ES cells (*Galiová et al., 2008*), and haplo-insufficiency of LMAC can limit certain lineages (*Sehgal et al., 2013*). In C2C12 cells, a myogenic cell line, LMAC becomes heavily concentrated at the nuclear periphery with myogenesis (*Markiewicz et al., 2005*), and mutant isoforms of LMAC can block myogenic lineage specification (*Markiewicz et al., 2005*). In MSCs, LMAC knockdown promotes adipocyte differentiation (and decreases osteoblast differentiation) (*Akter et al., 2009*; *Swift et al., 2013*), while forced expression of the progerin mutant isoform of LMAC increases osteogenic differentiation (and reduces adipogenic differentiation) (*Scaffidi and Misteli, 2008*).

Given the centrality of the nucleus in mediating both the transcriptional activities that define lineage specification, as well its role in mediating mechanical force transduction, the objective of this study was to investigate the manner in which both cellular and nuclear biophysical properties change during differentiation and how such changes might impact the mechanobiology of the cell. Specifically, we examined the nuclear mechanics and mechanosensitivity of MSCs as they transited toward a differentiated (fibrochondrogenic) lineage on aligned nanofibrous scaffolds. These scaffolds not only direct new tissue formation for fibrous tissue engineering (*Nerurkar et al., 2009*; *Baker et al., 2012*) but also provide a useful template for the application of coordinated cell deformation (*Nathan et al., 2011*; *Heo et al., 2011*; *Han et al., 2013*). Using this experimental platform, we show that MSC fibrochondrogenesis mediated by soluble factor addition over the long term (days to weeks) increases nuclear stiffness, alters organization of key nuclear structural elements, and as a consequence, changes the manner in which MSCs respond in the short term (seconds to minutes) to dynamic mechanical signals from the microenvironment. We further demonstrate that the repeated application of this short term dynamic mechanical stimulation (as opposed to application of soluble differentiation factors) can itself evoke some of these same changes in nuclear structure and mechanics, and that it does so more rapidly than soluble factor addition alone. This implies that nuclear

stiffening may be both a consequence and mediator of differentiation, tuning how progenitor cells interpret mechanical cues from their microenvironment. Such changes may act to first inform, and then reinforce, signals that promote and preserve lineage specification.

## Results

### Differentiation attenuates stretch-induced nuclear deformation

When bovine MSCs were exposed to differentiation media (Diff), expression of fibrochondrogenic markers (aggrecan and collagen II) increased (*Figure 1—figure supplement 1a, b*), consistent with previous findings (*Baker and Mauck, 2007*). To assess mechanical changes in the nucleus with differentiation, MSCs cultured on aligned scaffolds were subjected to 10% tensile stretch of the scaffold (as in [*Nathan et al., 2011*; *Heo et al., 2011*]) on specific days over the first 10 days of culture. This terminal assay provides an indirect measure of nuclear stiffness/deformability in the context of a living cell. At early time points, stretch of the scaffold resulted in both cell and nuclear deformation, as evidenced by a marked increase in the absolute nuclear aspect ratio (NAR) values in both control (Ctrl) and differentiation (Diff) conditions (*Figure 1a, c*). From day 5 onward, however, there was no change in the NAR in differentiated MSCs (Diff) with scaffold stretch, while nuclear deformation persisted in undifferentiated MSCs (Ctrl, *Figure 1b, c*).

Since this shift in the average response could simply represent a change in a subset of nuclei within the heterogeneous population, we next analyzed the distribution of NAR in all cells as a function of stretch and differentiation. On day 1, quantification showed a shift to a higher NAR with scaffold stretch for both groups (*Figure 1d*), but by day 7, differentiated cells (Diff) showed no shift in population NAR with scaffold deformation (*Figure 1e*). A similar effect, namely a reduction in nuclear deformation in response to stretch under differentiating conditions (Diff), was observed in both human bone marrow derived MSCs (hMSCs) (*Figure 1—figure supplement 2a–c*) and in a human embryonic stem (ES) cell line (*Figure 1—figure supplement 2d–e*) treated similarly.

### Altered nuclear mechanics in differentiated MSCs attenuates stretch-induced nuclear deformation

To distinguish between potential factors that may contribute to this attenuation of stretch-induced nuclear deformation, we carried out a series of experiments to measure nuclear stiffness, investigate the role of deposited matrix (*Figure 2a*), and queried the patency and contractility of the cytoskeletal network with differentiation (*Figure 2—figure supplement 1*).

First, peri-nuclear elastic modulus was measured by atomic force microscopy of cells removed from scaffolds and plated on tissue culture plastic. Stiffness was measured in both undifferentiated and differentiated MSCs (*Figure 2b, c*). In previous work (*Nathan et al., 2011*; *Driscoll et al., 2015*), we established that nuclear deformation with scaffold stretch was mediated primarily by the tensed actin cytoskeleton and its connections to the nucleus through nesprin-1 giant. Therefore, the peri-nuclear stiffness was measured both with and without the addition of cytochalasin D to remove potential contributions from the actin cytoskeleton (*Figure 2b, c*). Over seven days, the peri-nuclear stiffness did not change for undifferentiated MSCs (Ctrl), but increased by >50% in differentiated MSCs (Diff, *Figure 2d*). Treatment with CytoD (and DMSO vehicle) did not alter these values, indicating that stiffening was not due to changes in f-actin organization or mechanics, although this does not rule out potential contributions from other cytoskeletal elements (i.e. microtubules, and intermediate filaments).

Given that MSC differentiation culminates in abundant matrix production, which may shield the cell from applied stress, we next investigated the potential role of this deposited extracellular matrix on our observation of a loss of nuclear deformation with stretch. To do so, MSCs were removed from the scaffold on day 7 (with trypsin) and re-seeded for 24 hr onto fresh scaffolds. Re-seeded MSCs from both Ctrl and Diff conditions adopted a highly aligned morphology (*Figure 2e*). Consistent with the above findings, nuclei of these re-seeded differentiated MSCs (Diff) did not deform (*Figure 2f, g*) when the scaffold was stretched to 10%, while re-seeded undifferentiated MSC nuclei (Ctrl) increased in NAR by ~25%. These findings indicate that deposited extracellular matrix is not responsible for the observed attenuation in nuclear deformation with scaffold stretch. Additionally, we measured substantial increases in cytoskeletal contractility (which would tend to increase rather

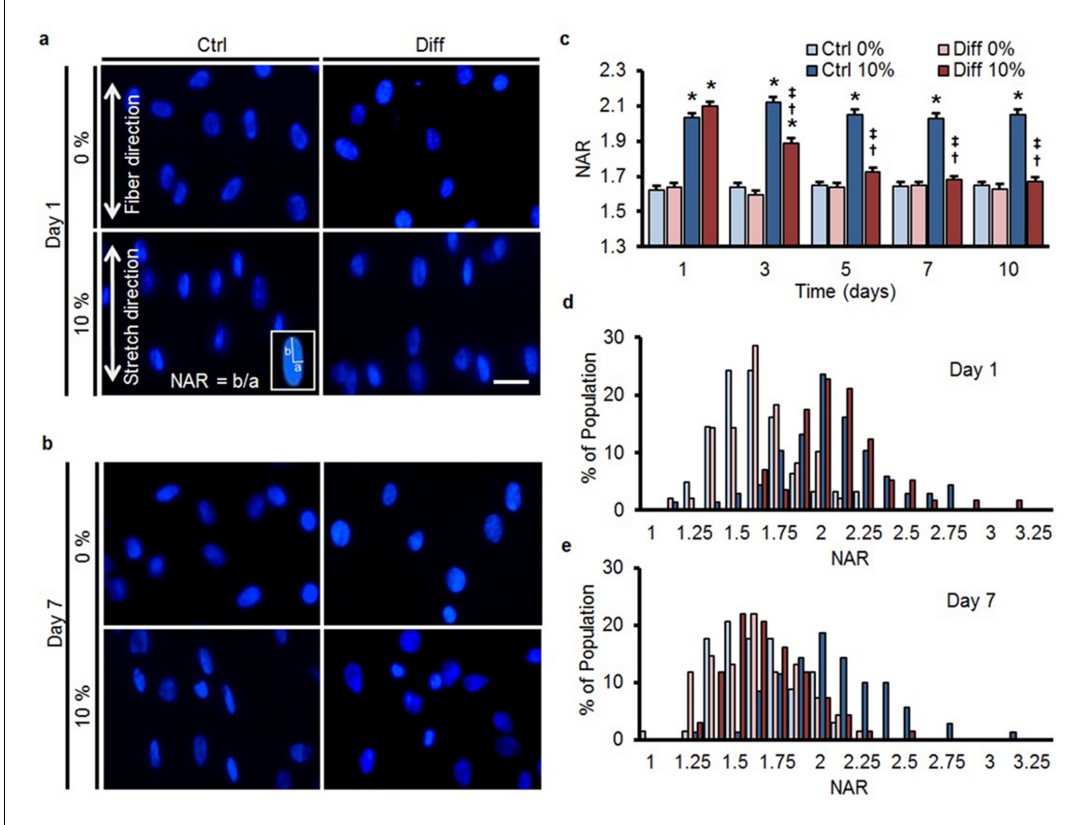

**Figure 1.** MSC differentiation reduces nuclear deformation with applied stretch. (a) On day 1, bovine MSCs on aligned nanofibrous scaffolds have elongated nuclei that are oriented in the prevailing fiber direction in both Ctrl and Diff conditions. Application of 10% scaffold stretch increased nuclear deformation under both culture conditions (bottom panels). (b) By day 7, application of 10% stretch continued to increase nuclear deformation in Ctrl conditions, while in Diff conditions nuclear deformation was almost completely absent. Scale: 20 μm. (c) Quantification of nuclear morphology showed that 10% stretch significantly increased MSC nuclear aspect ratio (NAR) in Ctrl conditions, irrespective of culture duration. Conversely, for MSCs in Diff conditions, a progressive decrease in the change in NAR with 10% stretch was observed with increasing culture duration (ANOVA, *p<0.05 vs. 0%, †p<0.05 vs. Ctrl, ‡p<0.05 vs. day 1, n = 102 nuclei per time point from 4–5 scaffolds per condition, absolute NAR values, mean ± SEM, experiments were carried out at least three times in full). Population distribution of MSC NAR in Ctrl and Diff conditions on day 1 (d) and day 7 (e); on day 1, 10% stretch shifted the distribution to the right for both conditions, while the shift was only apparent in Ctrl conditions on day 7. See *Figure 1—source data 1*.

The following source data and figure supplements are available for figure 1:

**Source data 1.** Changes in nuclear aspect ratio and population distribution with stretch.

**Figure supplement 1.** Type II collagen and aggrecan gene expression in bovine MSCs as a function of media condition and time.

**Figure supplement 2.** Differentiation-mediated reduction in strain transfer to the nucleus in human bone-marrow-derived MSCs and a human ES-cell line.

than decrease nuclear deformation, *Figure 2—figure supplement 1*). To interrogate the force generating and strain transfer capacity of the actin cytoskeletal network, MSCs were also cultured on elastomeric micropost array detectors following culture on aligned scaffolds for 7 days in Ctrl or Diff conditions. Traction force increased over the first 16 hr of exposure to TGF-β3 (*Figure 2—figure supplement 1a*). Actin stress fibers were prominent in both undifferentiated (Ctrl) and differentiated (Diff) cells, although differentiated cells induced significantly greater post deflections (*Figure 2—figure supplement 1b*). Quantification of the total strain energy per cell revealed a significant increase in traction forces (~5 fold) with differentiation (*Figure 2—figure supplement 1c*). On scaffolds, blockade of acto-myosin contractility decreased nuclear deformation in undifferentiated cells (Ctrl/Y27, *Figure 2—figure supplement 1d*), implicating the need for a patent and contractile

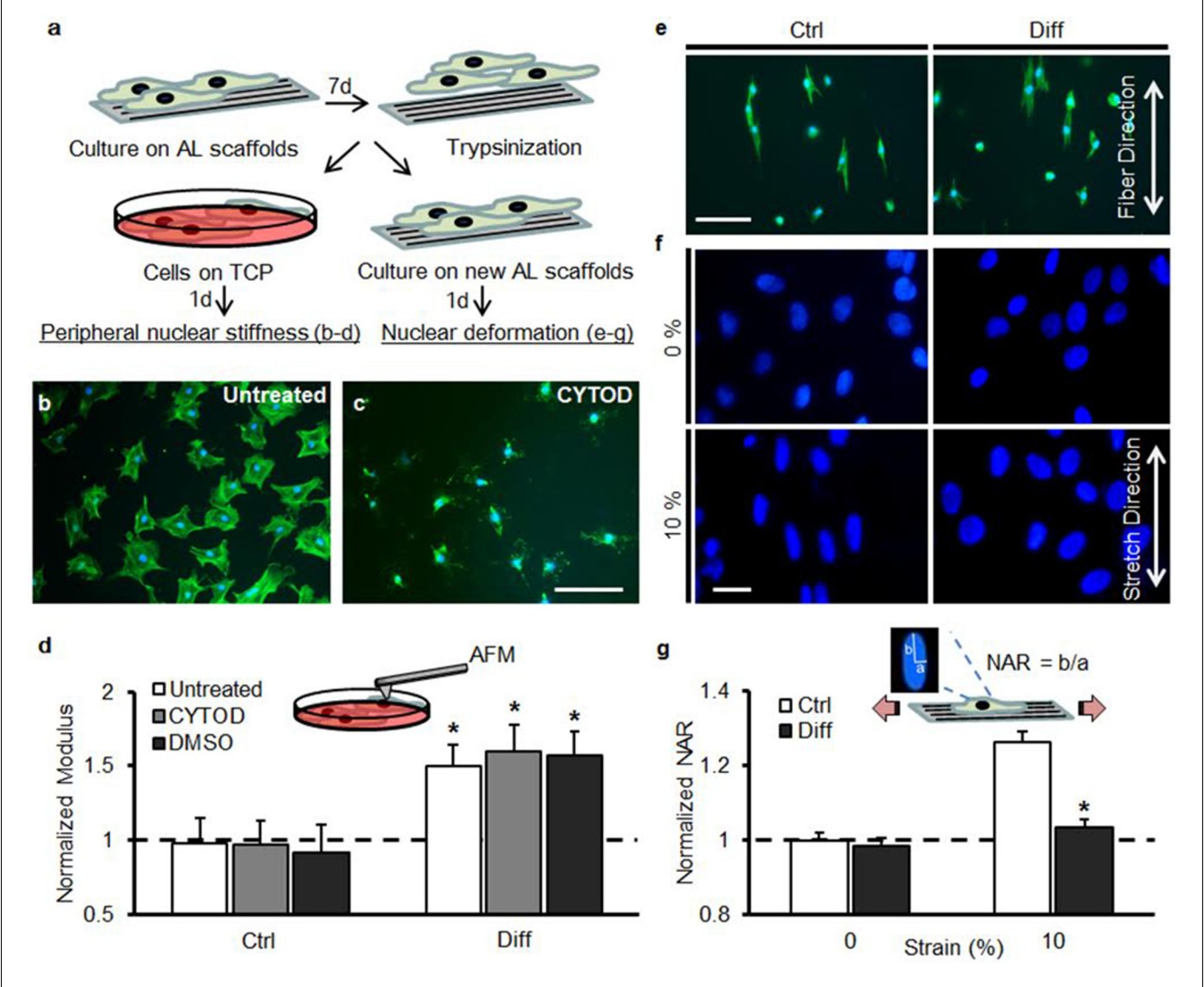

**Figure 2.** Decreased strain transfer to the nucleus with differentiation is a consequence of nuclear stiffening and not extracellular matrix deposition. (a) Schematic illustrating how MSCs were cultured on aligned (AL) scaffolds in Ctrl condition or Diff condition for 7 days followed by re-seeding on fresh scaffolds (for stretch studies) or re-plating on tissue culture plastic (TCP, for analysis of peri-nuclear stiffness by AFM). (b) Fluorescent staining of F-actin (green) in MSCs re-plated on TCP for one day and (c) disruption of F-actin via treatment with cytochalasin D (CYTOD) for 30 min. Scale: 100 µm. (d) The peri-nuclear stiffness of differentiated Diff/MSCs was significantly higher than that of undifferentiated Ctrl/MSCs after re-plating on day 7 (normalized to Ctrl values, ANOVA, *p<0.05 vs. Ctrl, n = 10 per condition, dashed line indicates peri-nuclear modulus on day 1, mean ± SD, experiments were carried out in duplicate). Treatment with CYTOD (or DMSO vehicle control) did not alter the measured peri-nuclear stiffness. (e) F-actin (green) and DAPI (blue) staining of MSCs re-seeded onto fresh scaffolds. Scale: 100 µm. (f) DAPI staining of nuclear morphology before (0%) and after (10%) scaffold stretch showed that differentiated MSC nuclei did not deform after transfer to a fresh scaffold. Scale: 20 µm. (g) Quantification of NAR confirmed lack of deformation of Diff nuclei (ANOVA, *p<0.05 vs Ctrl/10% after re-seeding, n = 50 nuclei/condition, mean ± SEM, experiments were carried out at least three times). See *Figure 2—source data 1*.

The following source data and figure supplement are available for figure 2:

**Source data 1.** Peri-nuclear stiffness measurements and changes in nuclear aspect ratio with stretch.

**Figure supplement 1.** Contractility increases in differentiated MSCs and is necessary for strain transfer to the nucleus.

cytoskeleton in the transfer of deformation from the scaffold to the nucleus. Taken together, these findings suggest that nuclear stiffening with differentiation dominates the response, resulting in a nucleus that no longer deforms when the cell is stretched.

## Structural reorganization and condensation of the nucleus with MSC differentiation

To understand the origin of these increases in nuclear stiffness with MSC differentiation, we next interrogated the amount and distribution of nuclear structural proteins, as well as the degree to which chromatin condensed under differentiating conditions. On day 1, LMAC was distributed throughout the MSC nucleus (*Figure 3a*), and in Ctrl conditions, this distribution did not change through day 7 (*Figure 3a* and *Video 1*). Under Diff conditions, LMAC remodeled to localize solely to the nuclear periphery (*Figure 3a, b* and *Video 2*). Localization to the nuclear periphery was accompanied by a ~50% increase in A-type lamin protein content (*Figure 3c, d*). Likewise, staining for H3K27me3 (a heterochromatin marker) revealed a marked increase in the number of heterochromatin-rich nodules throughout the nucleus with differentiation (Diff, p<0.05 vs. Ctrl, *Figure 3e, f*).

## Altered nuclear mechanics modulates mechanosensitivity of differentiated MSCs

The above findings suggest that the nuclei of differentiated cells transform from being a 'strain sink' (an object readily deformed along with the rest of the cell) to a 'stress concentrator' (an object that no longer deforms, and as a consequence forces other elements of the cell to deform to a greater extent). Given this transformation, we next asked whether such changes in nuclear stiffness alter the degree to which mechanical perturbation of the whole cell regulates signal transduction as a function of differentiation state. To answer this question, cells on scaffolds were stretched rapidly, and short-term changes in calcium signaling were used as an early readout of altered mechanotransduction. Additionally, the stiff nuclei of differentiated MSCs (Diff) were softened via chemical pre-treatment with the histone deacetylase (HDAC) inhibitor, trichostatin A (TSA [*Gerlitz and Bustin, 2010*]) and the soft nuclei of undifferentiated cells (Ctrl) were transiently stiffened by short-term osmotic shock using D-mannitol (DM) (*Irianto et al., 2013*) prior to the application of this mechanical perturbation.

Treatment of differentiated cells with TSA for 24 hr induces chromatin relaxation and, in this study, resulted in a marked reduction in both heterochromatin content and nuclear stiffness (Diff/TSA, *Figure 4—figure supplement 1a, b* and *Figure 4—figure supplement 1f*). While TSA did not alter the distribution or the amount of LMAC in these differentiated cells (*Figure 4—figure supplement 1c–e*), it did partially restore nuclear deformation in response to scaffold stretch (*Figure 4a, b*). To examine the link between nuclear stiffness and cellular mechanotransduction, we next evaluated changes in intracellular calcium in response to a rapidly applied stretch. In undifferentiated MSCs (Ctrl), a rapid stretch resulted in a small increase in intracellular $Ca^{2+}$. When the same perturbation was applied to differentiated MSCs (Diff), a much larger increase was observed (p<0.01, *Figure 4c, i*). Interestingly, this enhanced sensitivity was reversed by pre-treatment of differentiated MSCs with TSA (Diff/TSA). This sensitization to mechanical perturbation with differentiation (and blockade with TSA treatment) was also observed for aggrecan and collagen II gene expression (*Figure 4d, e*). With TSA pre-treatment (Diff/TSA), we found no significant changes in cytoskeletal structure, contractility (as assessed by myosin light chain phosphorylation, pMLCK) or migration rate (*Figure 4—figure supplement 2a–e*).

To test the converse, undifferentiated MSCs were transiently exposed to a hypertonic solution [500 mM D-mannitol (DM), (*Irianto et al., 2013*), Ctrl/DM] for 30 min, which has been shown to induce the rapid condensation of nuclear chromatin. This treatment resulted in MSC nuclei that were more resistant to deformation when the scaffold was deformed (*Figure 4—figure supplement 2f–g*, *Figure 4f–g*). Importantly, this treatment (Ctrl/DM) did not alter the organization of the actin cytoskeleton (*Figure 4—figure supplement 2h*). When stretch was applied to MSCs whose nuclei were artificially stiffer, but still undifferentiated, a marked increase in calcium mobilization was once again observed (*Figure 4h, i*).

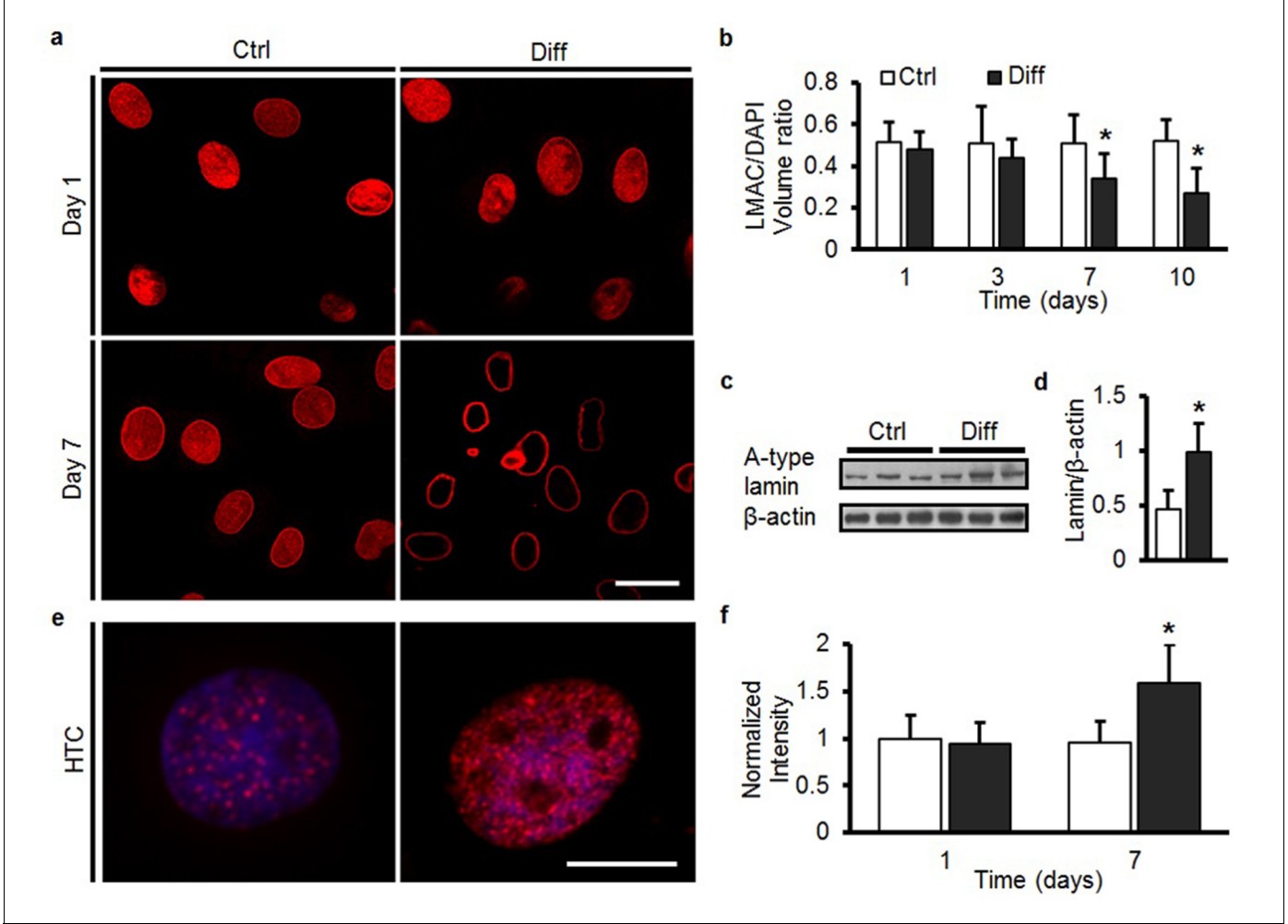

**Figure 3.** MSC differentiation results in marked nuclear reorganization. (**a**) On day 1, Lamin A/C (LMAC) was spread diffusely throughout the nucleus in both Ctrl and Diff conditions. By day 7, Diff conditions resulted in a restriction of LMAC to the nuclear periphery (scale: 20 µm). (**b**) Quantification showed that the nuclear volume occupancy ratio of LMAC (relative to DAPI) did not change in Ctrl conditions, while this ratio decreased significantly in Diff conditions as a function of culture duration (ANOVA, *p<0.05 vs. Ctrl, n = 31, mean ± SD, experiments wer carried out in triplicate). (**c**) Protein levels of A-type lamin (and β-actin) under Ctrl and Diff conditions. (**d**) Quantification showed a significant increase in A-type lamin with differentiation (Student's t-test, *p<0.05 vs. Ctrl, n = 6 from two biological replicates, mean ± SD). (**e**) Staining for heterochromatin (HTC) in Ctrl and Diff conditions on day 7. Scale: 5 µm. (**f**) HTC staining intensity in Diff conditions was significantly higher than in Ctrl conditions on day 7 (ANOVA, *p<0.05 vs. Ctrl, n = 25 nuclei/condition, mean ± SD, experiments were carried out at least three times). See *Figure 3—source data 1*.

The following source data is available for figure 3:

**Source data 1.** The nuclear volume occupancy ratio of LMAC, protein levels of A-type lamin and β-actin, and HTC staining intensity.

## Dynamic loading alters MSC nuclear mechanics and mechano-perception

The previous experiments demonstrated that MSC differentiation, guided by exogenous soluble factors, altered nuclear mechanics and mechano-perception. However, cells in mechanically loaded tissues experience repeated bouts of mechanical stimulation that, over time, can summate to drive lineage specification and/or regulate tissue homeostasis. To determine whether such mechanical signals on their own could elicit similar changes as the addition of soluble factors, undifferentiated MSCs were exposed to dynamic tensile loading in the absence of exogenous TGF-β3. These studies showed that short periods of dynamic loading [1 hr of dynamic stretch (3%, 1 Hz)] that was applied each day for 5 days did not alter LMAC distribution (*Figure 5a, b*). However, when the same loading

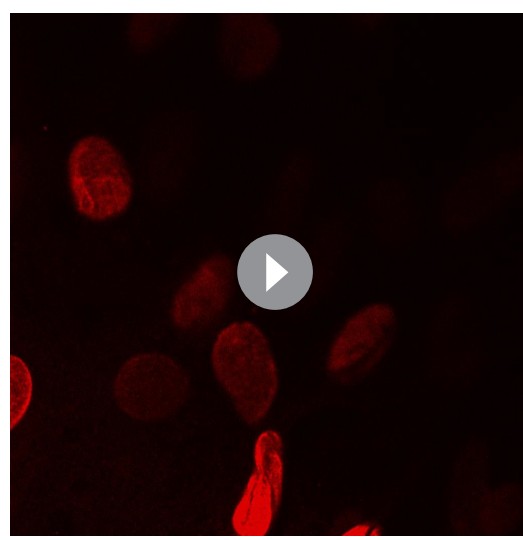

**Video 1.** 3D reconstruction of LMAC in the nucleus in undifferentiated bovine MSCs.

regimen was applied for 3 or 6 hr per day, LMAC reorganization was evident by day 5 (*Figure 5c, d*; *Figure 5—figure supplement 1a*). Strain magnitude also modulated this process; LMAC reorganization was evident with 1.5% and 3% stretch, but not at 0.75% stretch (applied for 6 hr per day, *Figure 5d–g*, *Figure 5—figure supplement 1b*). LMAC staining intensity and organization in these mechanically pre-conditioned MSCs (i.e. those exposed to higher levels of DL for longer periods of time) was similar to that observed in MSCs differentiated via soluble mediators (Diff, *Figure 5—figure supplement 1c*), suggesting that LMAC reorganization mediated through mechanical signals matches or exceeds that of soluble-factor-mediated differentiation. Similarly, dynamic loading (DL) conditions led to a marked increase in heterochromatin content, comparable to that induced by soluble factors (*Figure 5—figure supplement 1d, e*).

Given that the dynamic mechanical preconditioning above certain duration and magnitude elicited many of the same changes within the nucleus as soluble factor addition, we next investigated nuclear mechanics, nuclear deformability and response to mechanical perturbation in these cells. When mechanically pre-conditioned MSCs (DL, 3%, 1 Hz, 6 hr, 5 days of loading) were reseeded onto new scaffolds, nuclear deformation with 10% scaffold stretch was lower than for undifferentiated MSCs, approaching the levels observed in MSCs differentiated via soluble mediators (Diff, *Figure 5h*). Nuclei in DL pre-conditioned cells were stiffer than undifferentiated MSCs, although not as stiff as nuclei in MSCs differentiated via soluble factors (Diff, *Figure 5i*). Furthermore, DL increased aggrecan (AGG) and TGF-β gene expression to levels higher than in differentiation conditions (Diff, *Figure 5j, k*). The marked upregulation of TGF-β expression is interesting, given that it is a key factor included within differentiation medium. Indeed, addition of low levels of exogenous TGF-β3 (1 ng/mL) for 1 week resulted in changes in LMAC distribution and increases in heterochromatin in MSCs (*Figure 5—figure supplement 1f*). When a rapid stretch was applied to DL pre-conditioned MSCs, a greater increase in intracellular calcium was observed, compared to undifferentiated MSCs (ctrl, *Figure 5l*).

Overall, these data suggest that mechanical stimulation can drive lineage specification, including alterations in nuclear organization and mechanics, and that these changes may be mediated through autocrine/paracrine signaling. Importantly, in the absence of exogenous differentiation factors, dynamic loading alone can produce a mechanically-preconditioned nuclear state that is similar (but distinct) to that brought about by soluble factors.

## Dynamics of MSC nuclear remodeling in response to soluble factors and mechanical pre-conditioning

Our observation of nuclear remodeling after extended mechanical or biochemical stimulation

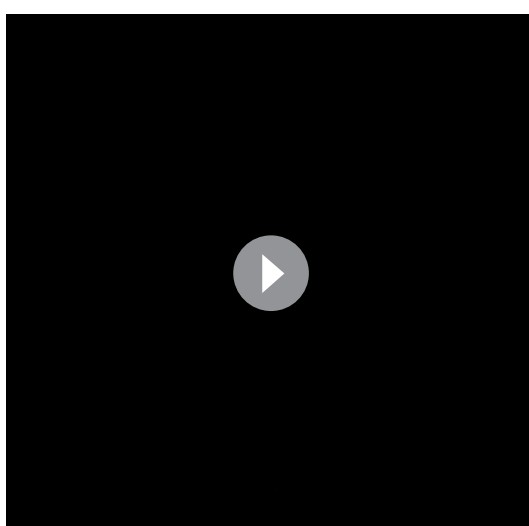

**Video 2.** 3D reconstruction of LMAC in the nucleus in differentiated bovine MSCs.

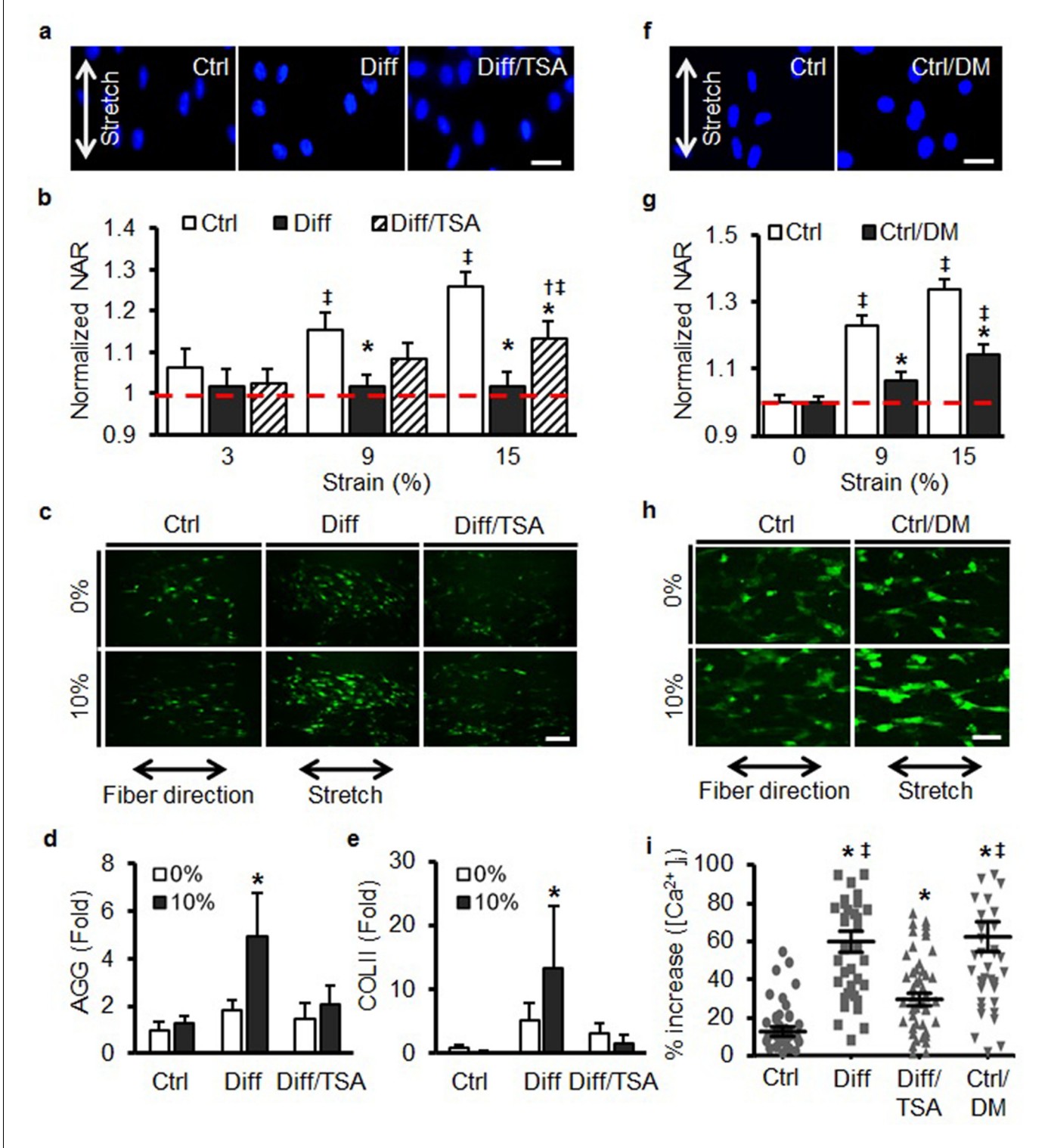

**Figure 4.** Increases in nuclear mechanics heighten mechanosensitivity of mesenchymal stem cells. (a) On day 7, 10% static stretch increased nuclear deformation in Ctrl conditions but did not alter nuclear shape under Diff conditions. With TSA treatment (Diff/TSA), nuclear elongation of differentiated MSCs was once again apparent with stretch. Scale: 20 μm (b) Quantification of NAR via tracking of individual nuclei. With TSA treatment of differentiated MSCs, nuclear deformation was once again observed with 15% stretch (normalized to Ctrl 0%, ANOVA, *p<0.05 vs. Ctrl, †p<0.05 vs. Diff, ‡p < 0.05 vs. Ctrl 0%, dashed line indicates Ctrl 0%, n = 28, mean ± SEM, experiments were carried out in triplicate). (c and i) Baseline intracellular $Ca^{2+}$ in MSCs was not different between groups. With 10% stretch, intracellular $Ca^{2+}$ showed slight increases in Ctrl conditions, and a much more marked increase in Diff conditions. TSA addition (Diff/TSA) reduced intracellular $Ca^{2+}$ release with stretch (Scale: 100 μm). (d) AGG and (e) COL II levels did not

*Figure 4 continued on next page*

*Figure 4 continued*

increase with stretch in Ctrl conditions, while significant increases were observed in Diff conditions. Addition of TSA (Diff/TSA), abrogated this stretch-induced response (10% static stretch for 1 hr, ANOVA, *p<0.05 vs. Ctrl, n = 3, mean ± SD, experiments were carried out in triplicate). (**f** and **g**) Pre-treatment of MSCs with D-mannitol (Ctrl/DM) decreased nuclear deformation with stretch (Scale: 20 μm, normalized to ctrl/0%, ANOVA, *p<0.05 vs. Ctrl, ‡p < 0.05 vs. Ctrl/0%, n = 40, dashed line indicates Ctrl 0%, mean ± SEM, experiments were carried out in triplicate). (**h** and **i**) Pre-treatment with D-mannitol (Ctrl/DM) increased intracellular calcium mobilization with stretch compared to Ctrl conditions (Scale: 50 μm, ANOVA, *p<0.05 vs. Ctrl, ‡p < 0.05 vs. Diff/TSA, n = 50, mean ± SEM, experiments were carried out in triplicate). See *Figure 4—source data 1*.

The following source data and figure supplements are available for figure 4:

**Source data 1.** Changes in nuclear aspect ratio and type II collagen and aggrecan gene expression with stretch.

**Figure supplement 1.** TSA treatment of differentiated MSCs softens nuclei and decreases heterochromatin, but does not alter Lamin A/C amount or distribution.

**Figure supplement 2.** TSA treatment of differentiated MSCs does not alter MLCK activity, actin structure, or migration speed.

(over a week time scale), led us to investigate the early remodeling of MSC nuclei in response to these cues. TGF-β3 (Diff) did not alter LAMC organization or gene expression over the first 3 days of culture (*Figure 6a, d*). However, within just 2 days under DL conditions, reorganization of LMAC was already apparent, with translocation to the nuclear envelope complete by day 3 (*Figure 6a*) and increased LMAC gene expression with DL at each time point (*Figure 6d*). Interestingly, HTC staining intensified more rapidly under DL conditions compared to Diff, with a higher number of heterochromatin-rich nodules present as early as day 1 (compared to day 3 under Diff conditions, *Figure 6b, c*). This precocious chromatin restructuring under DL conditions was further quantified using an image-based technique (*Irianto et al., 2013*) that determines a 'chromatin condensation parameter' (CCP) based on the amount of free space and number of visible edges in DAPI-stained images of the nucleus. This quantitative analysis showed a more rapid structural reorganization of the nucleus under DL conditions, compared to culture in differentiation medium (*Figure 6e, f*). Along with this, gene expression analysis revealed an early and robust response to dynamic loading in comparison to the addition of exogenous TGF-β3 (Diff, *Figure 6—figure supplement 1a–d*).

## Discussion

In addition to being the site for storage of genetic material and gene transcription, the nucleus, which is generally the stiffest element of all eukaryotic cells (*Dahl et al., 2008*), plays a central role in mechanical strain transduction from the external microenvironment. While the mechanisms by which nuclear deformation regulates transcriptional activities are only now beginning to be elucidated, our data suggest that the nucleus acts as a mechanical rheostat within progenitor cells, where increases in nuclear stiffness sensitize the entire transduction machinery to prime the cell response to mechanical perturbation. Our data show that soluble differentiation factors as well as mechanical perturbation on its own, elicit marked changes in nuclear deformability, stiffness and organization as a direct consequence of chromatin condensation and restructuring of the Lamin A/C network in both adult progenitor cells and in embryonic stem cells. This change in nuclear mechanics and organization led to a greater mobilization of intracellular calcium when pre-conditioned mesenchymal stem cells were exposed to stretch, likely as a consequence of the nucleus transforming from a strain sink (an object that readily deforms) to a strain concentrator (an object that stops deforming, forcing other components to deform more). These changes would be expected to increase stress and/or deformation occurring at nucleo-cytoskeletal and cytoskeletal linkages (i.e. LINC complex) to the ECM (i.e. focal adhesions). When one element of an elastic mechanical linkage stiffens, the other weaker elements must deform more when the system is elongated by a set amount. Indeed, the enhanced mechanical sensitivity seen in differentiated cells was reversed when the 'stiff' differentiated nucleus was softened and was enhanced when the 'soft' undifferentiated nucleus was stiffened through pharmacologic treatment.

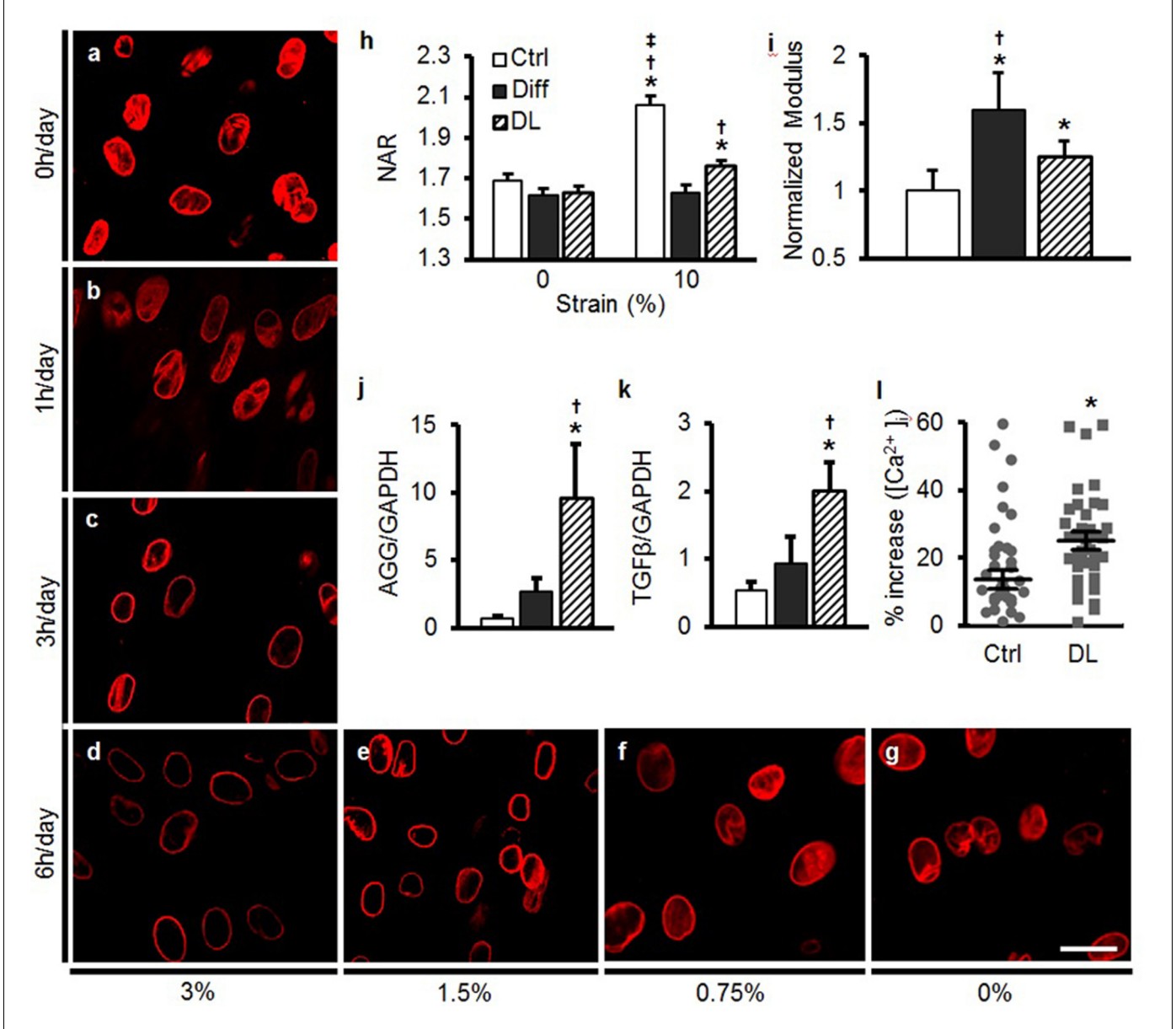

**Figure 5.** Dynamic loading induces nuclear reorganization and increases nuclear mechanics in undifferentiated MSCs. Dynamic loading (DL) for 5 days in the absence of soluble differentiation factors resulted in reorganization of LMAC in a manner dependent on the duration (0, 1, 3, or 6 hr/day, **a–d**) and magnitude (0, 0.75, 1.5, and 3% dynamic strain applied at 1 Hz, **d–g**) of applied loading (Scale: 20 μm). (**h**) DL (6 hr, 3%) decreased changes in NAR with 10% static stretch (ANOVA, *p<0.05 vs. 0%, †p<0.05 vs. Diff, ‡p<0.05 vs. DL, n = 52, mean ± SEM, experiments were carried out in triplicate). (**i**) DL nuclei were stiffer than those in undifferentiated MSCs, but were not as stiff as nuclei in MSCs differentiated (Diff) through soluble factor addition (ANOVA, *p<0.05 vs. Ctrl, †p<0.05 vs. DL, n = 10, mean ± SD, experiments were carried out in duplicate). (**j** and **k**) DL increased expression of AGG and TGF-β to levels exceeding those of differentiated MSCs (ANOVA, *p<0.05 vs. Ctrl, †p<0.05 vs. Diff, n = 5, mean ± SD, experiments were carried out in triplicate). (**l**) A rapid 10% stretch applied to DL-conditioned MSCs resulted in higher intracellular calcium release compared to the same perturbation of Diff MSCs (Student's t-test, *p<0.05, n =~40, mean ± SEM, experiments were carried out in duplicate). See *Figure 5—source data 1* .

The following source data and figure supplement are available for figure 5:

**Source data 1.** Changes in nuclear aspect ratio with stretch, peri-nuclear stiffness, and gene expression with the application of DL.

**Figure supplement 1.** Dynamic loading of undifferentiated MSCs induces nuclear reorganization comparable to differentiation with soluble factors.

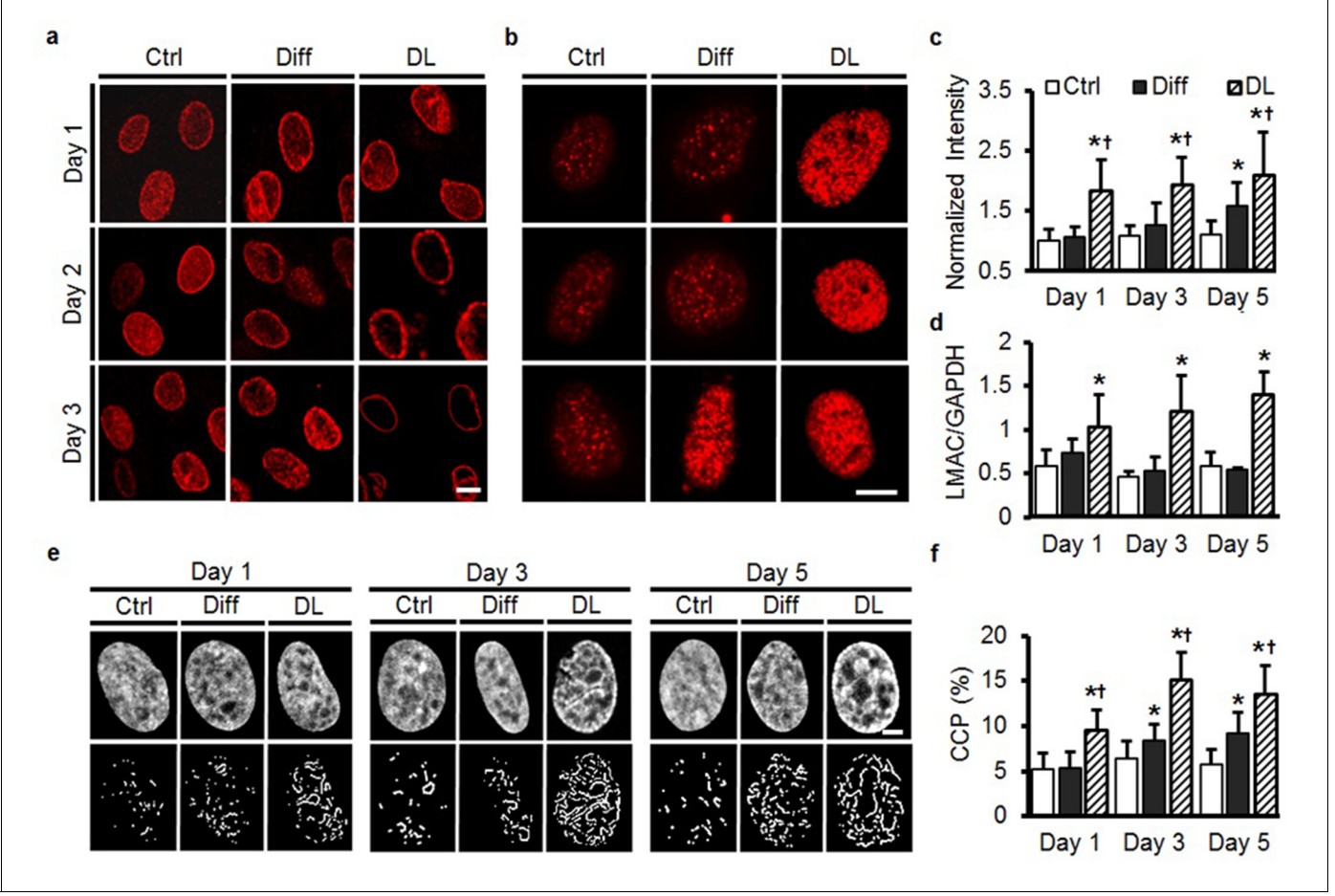

**Figure 6.** Rapid alterations in MSC nuclear architecture with dynamic loading. (**a**) Staining for LMAC showed little change in Ctrl conditions, and only subtle changes in Diff conditions through day 3. Conversely, DL promoted LMAC reorganization as early as day 2 (Scale: 5 µm). (**b**) Heterochromatin staining was evident in Ctrl conditions on day 3, but was already present on day one in DL conditions (Scale: 5 µm). (**c**) Quantification of heterochromatin staining showed significant differences in the DL group as early as day 1 (ANOVA, *p<0.05 vs. Ctrl, †p<0.05 vs. Diff, n = 35, mean ± SD, experiments were carried out in duplicate). (**d**) DL increased LMAC gene expression at each time point (ANOVA, *p<0.05 vs. Ctrl conditions, n = 3. Mean ± SD, experiments were carried out in triplicate). (**e**) Representative DAPI stained nuclei on days 1 through 5 with treatment (top row), and corresponding edge detection for CCP calculation (bottom row). Scale: 2 µm. (**f**) Chromatin condensation parameter (CCP) as a function of treatment group and time (ANOVA, *p<0.05 vs. Ctrl, †p<0.05 vs. Diff, n = 29–31 cells per condition per time point, mean ± SD, experiments were carried out in duplicate). See *Figure 6-source data 1* and *Source code 1*.

The following source data and figure supplement are available for figure 6:

**Source data 1.** HTC staining intensity, LMAC gene expression and CCP calculation.

**Figure supplement 1.** Rapid changes in fibrocartilaginous gene (Sox9, BMP2, TGF-β, and Aggrecan) expression with dynamic loading.

This notion of the nucleus as a dynamic mechanical feature of the cell that is responsive to the microenvironment is consistent with and extends several recent findings. Swift and colleagues recently showed that the stiffness of native tissue extracellular matrices defines the resting levels and ratios of Lamin isoforms in adult cells, with stiffer microenvironments increasing nuclear mechanics (*Swift et al., 2013*). Our data extend this concept to cells transiting from an undifferentiated to a differentiated phenotype as a result of soluble factor addition and exposure to repeated mechanical deformtion. This phenomenon may drive cell differentiation or maintain the differentiated phenotype to better match the dynamic mechanical tissue environment in which they operate. For instance, as soon as muscular contraction begins in the developing embryo, mesenchymal precursors are

exposed to dynamic mechanical perturbation, and blockade of mechanical signaling during development interrupts joint formation and tissue maturation. In the context of injury and repair, recruited stem cells experience the mechanical environment of the tissue being repaired, and so for musculo-skeletal tissues, experience dynamic mechanical stimulation as part of their normal microenvironment. Thus, dynamic mechanical forces transmitted through the environment may continuously adjust nuclear mechanics to modulate progenitor cell mechanosensitivity during tissue formation and repair.

The mechanism through which soluble factor addition or mechanical perturbation results in nuclear reorganization is not wholly elucidated, but these inputs may share some common downstream pathways. For example, both BMP and TGF-β signaling increase baseline cellular contractility (*Bhadriraju et al., 2007*). Our data show that TGF-β expression is highly upregulated with dynamic stretch at early time points and while TGF-β in the media did not reach detectable levels, we did observe increased Smad3 signaling in response to stretch (*Figure 6—figure supplement 1e*, [*Furumatsu et al., 2013*; *Li et al., 2015*]). Thus, elements of the TGF-signaling pathway may be mechano-responsive, in a ligand or ligand-less fashion (*Heo et al., 2016*). Mechanical perturbation itself can also increase cell contractility and spreading (*Cui et al., 2015*), and influence the subsequent mechano-response. Kaunas et al. showed that artificially increasing Rho GTPase activity in endothelial cells sensitizes their response to applied stretch (*Kaunas et al., 2005*), and Deguchi et al. showed that fluid shear stress, which also increases contractility, can stiffen the endothelial cell nucleus (*Deguchi et al., 2005*). Recent work by the Shivashankar and Burridge groups has shown that local application of mechanical force to the apical surface of cells causes local chromatin reorganization (*Iyer et al., 2012*) and that pulling on isolated nuclei results in changes in Lamin organization and nuclear stiffening (*Guilluy et al., 2014*).

Regardless of the path to nuclear reorganization, our data provide evidence that the stiffer nucleus of differentiated cells can act as a stress concentrator (*Figure 7*), sensitizing these cells to mobilize calcium to a greater extent when external deformation is applied. This increased calcium mobilization may itself impact lamin processing (*Kalinowski et al., 2013*), creating a feed-forward loop to ensure that the nucleus remains patent under conditions of high mechanical activity. We recently showed that ATP-purinergic signaling is a central modulator in the load-induced-chromatin condensation in MSCs, as is signaling thorough Smad proteins (*Heo et al., 2015*, *2016*). Further, we have established that the giant isoform of nesprin-1 (a component of the LINC complex that mediates actin connectivity to the nucleus) is essential for mechanical force transfer to the nucleus and activation of the YAP/TAZ pathway in response to exogenous mechanical loading (*Driscoll et al., 2015*). These signaling pathways may likewise impact lamin processing and chromatin condensation.

While both mechanical forces and soluble factors had similar effects on nuclear properties and reorganization, those mediated by soluble factor addition arose more slowly than those wrought by mechanical forces. This suggests that the mechanical environment plays a significant role in dynamically modulating nuclear properties. Given that such changes act to first inform, and then reinforce, signals that promote and preserve lineage specification, this time dependence has implication for therapeutic use. That is, as MSCs transit between phenotypes, one would expect that their current differentiation status and time history of exposure to soluble or mechanical factors could influence their mechanobiologic response upon implantation. This understanding may aid in the design of biomaterials that can promote (or restrict) certain lineages based on nuclear remodeling. For example, if nuclear stiffening via Lamin reorganization is counter the intended differentiation pathway, then materials may be modified with ligands that interrupt mechano-sensing pathways, for example by decreasing cell contractility based on ligand presentation (*Cosgrove et al., 2016*). Alternatively, materials could be engineered to promote nuclear deformation (e.g. by aligning the cell population from the outset) so as to more efficiently transfer strain from the material to the nucleus to promote nuclear remodeling and achieve a differentiated status more quickly (*Li et al., 2011*).

In addition to material design, we also noted a path-dependent aspect in terms of the end results of nuclear reorganization; soluble factors resulted in complete abrogation of nuclear deformation, while mechanically mediated reorganization of the nucleus resulted in only a diminution of nuclear deformation with stretch. If nuclear deformation is essential for physiologic function, then complete elimination of nuclear deformation (as a consequence of soluble factor addition) may result in aberrant mechano-signaling. This has implications for establishing better cellular biophysical benchmarks by which small molecule drivers of differentiation are judged. These benchmarks may prove relevant

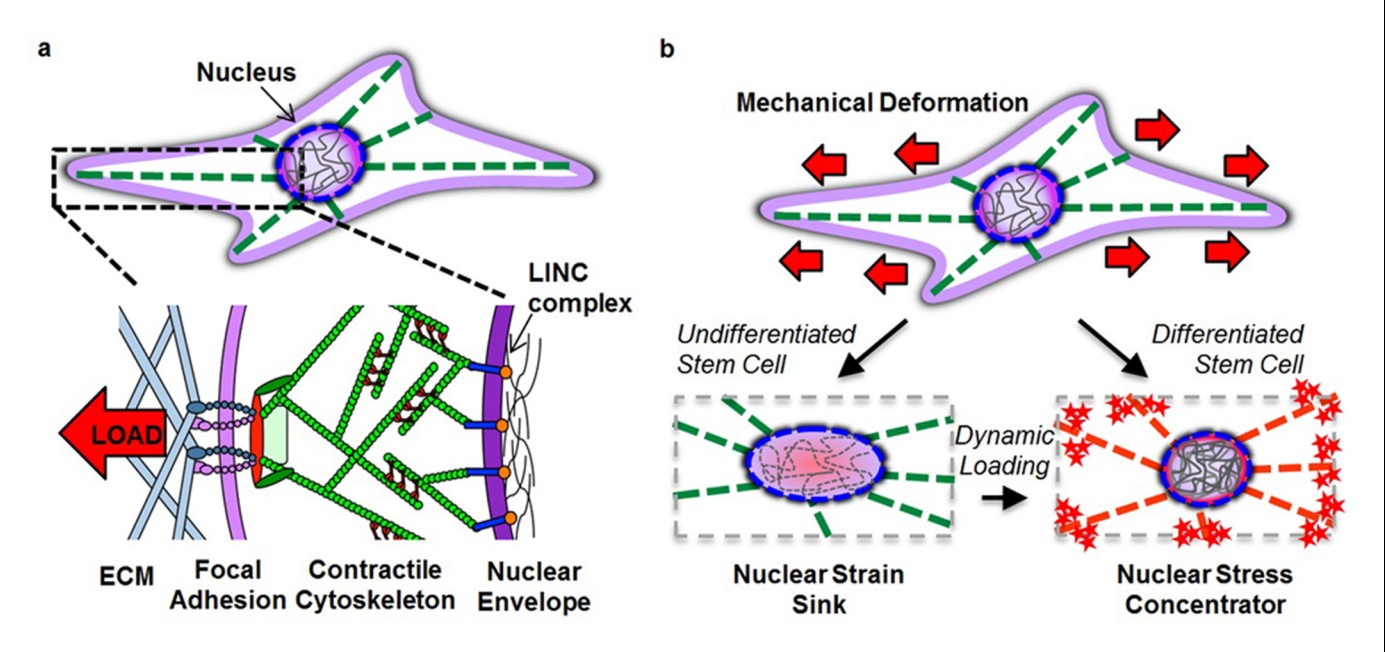

**Figure 7.** Altered nuclear mechanobiology with stem cell differentiation. (a) The nucleus is mechanically linked to the extracellular environment through linker of nucleus and cytoskeleton (LINC) complex connections to a contractile cytoskeleton that interact with the extracellular matrix (ECM) through focal adhesion complexes. (b) When progenitor cells differentiate, the nucleus transforms from a deformable 'strain sink' into a comparatively rigid (relative to the cytoskeleton) 'stress concentrator', with localization of Lamin A/C to the nuclear envelope (red circle in nucleus) and an increase in heterochromatin content (grey lines in nucleus). In the undifferentiated state, the nucleus deforms along with the cell, resulting in little added strain (or stress) in the cytoskeleton (green filaments). In the differentiated state, the lack of deformation of the nucleus concentrates deformation (and stress) in the cytoskeletal network (red filaments) and at its connections. This transformation hyper-sensitizes the differentiated cell to respond to mechanical perturbation (red stars) by increasing stress at each point of cytoskeletal connectivity (focal adhesions and LINC complex). Mechanical inputs (when repeated dynamically) can evoke similar changes in nuclear architecture and mechanics and increase cell mechanosensitivity to direct lineage specification in the absence of exogenous soluble differentiation factors.

for cell-based therapeutics, wherein naive stem cells are placed in a mechanically loaded environment. While it is not yet clear how permanent these changes are once soluble and mechanical stimuli are removed, it does suggest that care must be taken to preserve the appropriate mechanoresponse. In conclusion, the data presented herein demonstrates a mechanism whereby nuclear architecture and stiffness can be tuned to dynamically regulate cellular mechanosensitivity to promote and preserve lineage specification.

## Materials and methods

### Cell isolation, scaffold seeding and culture

Mesenchymal stem cells (MSCs), meniscal fibrochondrocytes (MFCs) and chondrocytes were isolated from juvenile bovine tibiofemoral joints (3–6 months old, Research 87, Inc., Boylston, MA) (*Baker and Mauck, 2007*; *Huang et al., 2010b*). The cells were cultured and expanded in a basal medium (BM) consisting of Dulbecco's minimal essential medium (DMEM) containing $1\times$ penicillin/streptomycin/fungizone and 10% fetal bovine serum. A 150 µl aliquot containing $2.5 \times 10^4$ or $2 \times 10^5$ cells was seeded on each side of the electrospun aligned nanofibrous scaffold, followed by incubation in BM at 37°C for 2 hr to allow for cell attachment. After attachment, scaffolds were transferred to a chemically defined serum-free medium (*Mauck et al., 2006*). The chemically defined serum-free medium consists of high-glucose DMEM with $1\times$ penicillin/streptomycin/fungizone, 0.1 µM dexamethasone, 50 µg/ml ascorbate 2-phosphate, 40 µg/ml l-proline, 100 µg/ml sodium pyruvate, 6.25 µg/ml insulin, 6.25 µg/ml transferrin, 6.25 ng/ml selenous acid, 1.25 mg/ml bovine serum albumin and 5.35 µg/ml linoleic acid (Life Technologies, NY, USA). Medium was further

supplemented with/without 10 ng/ml TGF-β3 (Ctrl/Diff, R&D Systems, Minneapolis, MN). Cell-seeded scaffolds were cultured in this medium for up to 10 days. Human MSCs were isolated from surgical waste tissue from patients undergoing total joint replacement at the University of Pennsylvania, with institutional approval. A human embryonic stem cell line was acquired from a commercial source (BioTime, Alameda, CA) and expanded according to the manufacturers guidelines (*Sternberg et al., 2012*).

## Fabrication of aligned nanofibrous scaffolds

Aligned poly(ε-caprolactone) (PCL) nanofibrous scaffolds ( ~0.50 mm thick) were fabricated by electrospinning onto a rotating mandrel. Briefly, PCL (Bright China Limited, China) was dissolved in a 1:1 solution of tetrahydrofuran and N, N-dimethylformamide (T427-1 and D119-1; Fisher Scientific, Pittsburgh, PA) at 14.3% w/v over 48 hr. The PCL solution was loaded into a 20-ml syringe tipped with an 18G stainless steel needle. This needle served as a 'spinneret' and was charged to +13 KV (ES30P-5W; Gamma High Voltage Research Inc., Ormand Beach, FL) relative to ground. The PCL solution was extruded at a rate of 2.5 ml/hr using a syringe pump (KD Scientific, Model 780100). Fibers were collected onto a grounded cylindrical mandrel rotating with a surface velocity of 10 m/s, with the mandrel positioned 15 cm from the spinneret. Aluminum shields charged to +5 kV were used to focus the fiber jet onto the collecting mandrel. Additional details on this scaffold fabrication process can be found in (*Heo et al., 2011*).

## Single cell measures of contractility with differentiation

To query cell contractility changes with differentiation, a custom micro-post array detector (mPAD) was utilized (*Fu et al., 2010*; *Yang et al., 2011*). After 7 days on scaffolds, MSCs were detached and re-seeded onto an array of elastomeric micro-pillars (spring constant: 18.19 nN/μm), and cultured for 1 day before traction force measurements were taken. Images of live single cells deflecting the underlying micro-pillars were acquired (n = 20/group) using a fluorescence microscope equipped with a temperature and $CO_2$-controlled incubator. Post deflections were visualized via DiI labeling of post tips and used to calculate strain energy per cell (*Yang et al., 2011*). Cells were fixed (4% PFA), permeabilized and stained with phalloidin and DAPI to visualize the actin cytoskeleton and nucleus, respectively.

## Assays of cell migration with differentiation

Migration of undifferentiated, differentiated and TSA-treated MSCs was evaluated using a standard 'scratch' assay. MSCs were released from scaffolds by trypsin treatment on day 7 and reseeded into 6-well tissue culture plates at a density of $2 \times 10^5$ cells per well in the same media in which they had been cultured (i.e. Ctrl or Diff conditions). After 1 day (to allow for attachment), confluent monolayers were scratched with a 2.5 μl pipette tip and washed three times with PBS to remove cell debris. In some Diff groups, TSA was added to Diff media 24 hr prior to wounding. Images of the wound area were taken at regular intervals using an inverted microscope and the degree of wound closure computed using Image J (RRID:SCR_003070). Wound area was normalized to the initial scratch area for each sample.

## Visualization of cell and nuclear structural proteins and organization

Cells on the nanofibrous scaffolds were fixed with 4% PFA at 37°C for 30 min and then immunostained for Lamin A/C (LMAC) using a mouse anti-Lamin A/C primary antibody (1:100, MA3-1000, Thermo, IL, RRID:AB_325377). Anti mouse IgG-TRITC (T2402, Sigma, St Louis, MO, RRID:AB_261618) was used as secondary antibody. Nuclei were counterstained with 4',6-diamidino-2-phenylindole (DAPI, Prolong Gold Reagent, P-36931, Life Technologies, NY). The spatial localization of LMAC was evaluated by confocal microscopy (Zeiss, LSM 510, Penn Biomedical Imaging Core). The ratio of LMAC nuclear occupancy was calculated from the volume of LMAC stain divided by total DAPI-stained nuclear volume using Volocity software (Volocity 5.0, Improvision Inc.). To assess chromatin condensation, nuclei on scaffolds were stained with DAPI, and scanned across their mid-section using a confocal microscope (Zeiss, LSM 510). To generate a chromatin condensation parameter (CCP), a gradient-based Sobel edge detection algorithm was employed using MATLAB to measure the edge density for individual nuclei (*Irianto et al., 2013*). Phalloidin-AlexaFluor 488 (Invitrogen,

Carlsbad, CA, RRID:AB_2315147) was used to stain the actin cytoskeleton. Heterochromatin (HTC) levels in MSC nuclei were assessed by the degree to which histone H3 was tri-methylated using the mouse anti-Histone H3 (tri methyl K27) antibody (ab6002, Abcam, MA, RRID:AB_305237). All images were collected using a fluorescent microscope (Zeiss, Axioplan-2 imaging fluorescent microscope) at 100× magnification, with HTC staining intensity quantified using MetaMorph (Molecular Devices Inc., CA).

## Perturbation of MSC nuclear stiffness and condensation

To decrease chromatin condensation in differentiated cells, MSCs that had been differentiated (Diff) for 6 days were treated with the histone deacetylase (HDAC) inhibitor Trichostatin A (TSA, T8552, Sigma-Aldrich, St. Louis, MO, USA) at 0.5 µg/ml for the final 24 hr of culture (in differentiation medium: Diff/TSA). To increase nuclear condensation in undifferentiated cells (Ctrl), MSCs on scaffold were exposed to a 500 mM solution of D-mannitol (Ctrl/DM) for 30 min as in (*Irianto et al., 2013*).

## Static and dynamic mechanical perturbation of cell-seeded scaffolds

Nuclear deformation of cells on aligned scaffolds was first quantified by applying a quasi-static tensile stretch to cell-seeded scaffolds ($2.5 \times 10^4$ cells per scaffold) using a custom device designed to permit visualization on an epi-fluorescent inverted microscope during stretch (*Nathan et al., 2011*). Cell-seeded scaffolds were stretched to 0% and 10% grip-to-grip strain at 2.5 %/min on days 1, 3, 5, 7 and 10. Constructs were fixed in the deformed state (in 4% paraformaldehyde) immediately after stretch, and nuclei were stained with DAPI after permeabilization with Triton X-100 (0.1%) for 10 min. All images were captured on an inverted fluorescent microscope (Nikon T30, Nikon Instruments, Melville, NY) equipped with a CCD camera. At each strain level, the nuclear aspect ratio defined as the long axis (b) divided by the perpendicular short axis (a) (NAR=b/a) was calculated from DAPI-stained nuclei using Image J. In subsequent studies, changes in NAR were tracked for individual MSC nuclei at each strain step (*Han et al., 2013*; *Driscoll et al., 2015*). In some studies, the Rho kinase inhibitor, Y27632, was included at 10µM overnight prior to and during scaffold stretch.

To determine the effects of cyclic tensile deformation applied over a longer period, MSC seeded scaffolds ($2 \times 10^5$ cells per side) were loaded into a custom bioreactor (*Baker et al., 2011*). Dynamic tensile loading (DL) was applied in Ctrl condition at 1 Hz for varying durations (up to 6 hr/day) and at various strain magnitudes (up to 3% strain) following 2 days of pre-culture in Ctrl media.

## Measurement of peri-nuclear stiffness by atomic force microscopy

MSC peri-nuclear stiffness was determined by atomic force microscopy (AFM, DAFM-2X, Veeco, Woodbury, NY). For this, cells were removed from scaffolds by trypsinization and re-plated onto tissue culture plastic for 24 hr. Cells were probed using a silicon nitride probe with a pyramidal tip (a spring constant of 0.06 N/m, DNP, Veeco) (*Solon et al., 2007*). To quantify peri-nuclear stiffness, the first 400 nm of tip deflection from the horizontal (Dd) were fit with the Hertz model modified for a conical tip (*Solon et al., 2007*). To confirm that AFM measurements represented changes in nuclear stiffness, rather than overall cell stiffness, control experiments were performed wherein the actin cytoskeleton was disrupted with 1 µM cytochalasin D (CYTOD, C2618, Sigma-Aldrich) for 30 mins prior to AFM probing. Control cells were treated with the carrier DMSO. Values of peri-nuclear stiffness were taken as the average of measurements made in three different regions of the nucleus.

## Quantification of cell and nuclear structural proteins

Semi-quantitative analysis of protein levels and phosphorylation states were assessed using Western blot analysis. For this, cell-seeded scaffolds were washed three times with PBS and lysed for 30 min on ice using radioimmunoprecipitation assay (RIPA) lysis buffer containing protease (p-8340, Sigma, St Louis, MO) and phosphatase inhibitors (p-2580, Sigma, St Louis, MO). Cell lysates were cleared by centrifugation at 10,000 g for 10 min, and protein concentrations were determined by Lowry assay (DC Protein Assay Kit, Biorad, CA). Cell lysates were separated on gradient poly-acrylamide gels (Mini-Protean TGX gels Any KD, 456–9034, Biorad, CA) in running buffer (Tris/Glycine/SDS Buffer, 161–0732, Biorad, CA). Separated proteins were transferred to nitrocellulose membranes

(iBlot Gel Transfer Stacks Nitrocellulose, Regular, IB3010-01, Invitrogen Life Technologies, Carlsbad, CA) and blocked for 1 hr in either 5% (w/v) bovine serum albumin (Sigma) or 5% (w/v) dry milk powder in PBS containing 0.1% (v/v) Tween 20 (PBST, Acros Organics, Geel, Belgium). The transferred membranes were probed with antibodies for A-type lamin (MA3-1000, Thermo, IL), phosphorylated myosin light chain kinase (pMLCK, LS-C25729, LSBio, RRID:AB_903557), pSmad3 (C25A9, Cell Signaling, RRID:AB_2193207) and β-actin (#4967, Cell Signaling, RRID:AB_2629465). Anti-rabbit IgG, HRP-linked Antibody (7074 s, Cell signaling, RRID:AB_657966) or HRP AffiniPure Goat Anti-Mouse IgG (H+L) (Catalog No.115-035-062, Jackson Immoresearch Laboratories, Inc., RRID:AB_2338504) were used as secondary antibodies. Blots were visualized by chemiluminescenc (34080, Thermo scientific, IL), and signal intensity was quantified by densitometry using ImageJ with normalization to β-actin.

## Analysis of gene expression by real-time PCR

Total RNA in constructs was isolated using Trizol reagent followed by phenol-chloroform separation. mRNA was quantified using a Nanodrop spectrometer (ND-1000, Nanodrop Technologies, Wilmington, DE), and cDNA synthesized using the SuperScript First Strand Synthesis kit (Invitrogen, Life Technologies, Carlsbad, CA). Amplification was carried out using an Applied Biosystems Step One Plus real-time PCR system, with intron spanning primers and Fast SYBR Green Reaction Mix (#4385617, Applied Biosystems, Foster City, CA). Expression of collagen type II (Col2), aggrecan (AGG), sox9, TGF-β (TGFβ), BMP-2, and LMAC were determined and normalized to the housekeeping gene glyceraldehyde-3-phosphate dehydrogenase (GAPDH) using the comparative Ct method.

## Real-time analysis of intracellular Ca$^{2+}$ levels with stretch

To determine whether MSC differentiation resulted in increased mechano-sensitivity, intracellular calcium levels were measured during scaffold stretch using a calcium-sensitive fluorescent indicator. On day 7, undifferentiated and differentiated MSCs were loaded with 25 mM fluo-4/acetoxymethyl ester (fluo-4AM; Molecular Probes, Eugene, OR) in 0.05% Pluronic F-127 (Molecular Probes, NY) and 1% HEPES (Gibco, NY) in control media. Scaffolds were incubated with dye solution for 30 min, followed by incubation in fresh control medium. Real-time images were obtained using an inverted fluorescence microscope (Nikon T30, Nikon Instruments, Melville, NY) during scaffold stretch. Additional samples were pre-incubated for 24 hr with TSA as above prior to application of scaffold stretch. To measure the calcium response after DL pre-conditioning, cells were seeded on scaffolds for 2 days and then dynamically loaded for 5 days (3% strain, 1 Hz, 6 hr/day). Afterwards, samples were stained with fluo-4AM dye and subjected to a static stretch. For the osmotic challenge study, cells were seeded on scaffold for 2 days, treated with D-mannitol (DM) for 30 min, stained with Cal-520AM (#21130, AAT Bioquest, Sunnyvale, CA) and then subjected to static stretch. A step static stretch was applied at a rate of 1 %/sec up to 10%. Individual MSCs were tracked during this deformation, and the signal intensity before and after stretch was measured using MetaMorph and normalized to background intensity. Intracellular Ca$^{2+}$ concentration in n > 50 cells was tracked across three scaffolds for each condition assayed.

## Statistical analyses

Statistical analysis was performed using Student's t-test or ANOVA with Fisher's LSD post hoc testing (SYSTAT v.10.2, Point Richmond, CA). Results are expressed as mean ± SEM or SD, as indicated in the figure legends. Differences were considered statistically significant at $p < 0.05$.

## Acknowledgements

This work was supported by the Human Frontiers in Science Program, the Penn Center for Musculoskeletal Disorders (P30 AR050950), the National Institutes of Health (R01 AR056624, R01 EB02425, R01 GM074048, and T32 AR007132), the RESBIO Technology Resource for Polymeric Biomaterials and a Marie Curie Intra European Fellowship (GENOMICDIFF 301509). BMB acknowledges financial support from a Ruth L Kirschstein National Research Service Award (EB014691) and NIH Pathway to Independence Award (K99HL124322). The authors would like to acknowledge technical contributions to this work from Dr Woojin Han.

# Additional information

## Funding

| Funder | Grant reference number | Author |
|---|---|---|
| National Institutes of Health | | Su-Jin Heo<br>Tristan P Driscoll<br>Nandan L Nerurkar<br>Brendon M Baker<br>Michael T Yang<br>Christopher S Chen<br>Robert L Mauck |
| Human Frontier Science Program | | Su-Jin Heo<br>Tristan P Driscoll<br>Stephen D Thorpe<br>David A Lee<br>Robert L Mauck |
| The Penn Center for Musculoskeletal Disorders | P30 AR050950 | Su-Jin Heo<br>Tristan P Driscoll<br>Robert L Mauck |
| Marie Curie Intra European Fellowship | GENOMICDIFF 301509 | Stephen D Thorpe |
| NIH Pathway to Independence Award | K99HL124322 | Brendon M Baker |
| National Research Service Award | EB014691 | Brendon M Baker |
| National Institutes of Health | R01 GM074048 | Christopher S Chen |
| National Institutes of Health | R01 AR056624 | Robert L Mauck |
| National Institutes of Health | R01 EB02425 | Robert L Mauck |
| National Institutes of Health | T32 AR007132 | Robert L Mauck |

The funders had no role in study design, data collection and interpretation, or the decision to submit the work for publication.

## Author contributions

S-JH, TPD, SDT, NLN, BMB, Conception and design, Acquisition of data, Analysis and interpretation of data, Drafting or revising the article; MTY, Conception and design, Acquisition of data, Analysis and interpretation of data; CSC, Analysis and interpretation of data, Drafting or revising the article; DAL, Conception and design, Analysis and interpretation of data, Drafting or revising the article; RLM, Conception and design, Analysis and interpretation of data, Drafting or revising the article, Contributed unpublished essential data or reagents

## Author ORCIDs

Stephen D Thorpe, http://orcid.org/0000-0002-4707-7756
Robert L Mauck, http://orcid.org/0000-0002-9537-603X

# Additional files

## Supplementary files

• Source code 1. MATLAB code for calculation of chromatin condensation parameter (CCP).

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
