## [Decision Letter]

Thank you for submitting your article "Differentiation Alters Stem Cell Nuclear Architecture, Mechanics, and Mechano-Sensitivity" for consideration by *eLife*. Your article has been reviewed by three peer reviewers, and the evaluation has been overseen by a Reviewing Editor and Fiona Watt as the Senior Editor. The reviewers have opted to remain anonymous.

The reviewers have discussed the reviews with one another and the Reviewing Editor has drafted this decision to help you prepare a revised submission.

1) The primary concern to address is the ability to discriminate differences in nuclear mechanics independent of differentiation and the large differences in time scales examined (see reviewer 2's comments below). Specifically, examination of potential differences in undifferentiated and differentiated cells both with and without TSA treatment should be performed.

2) Small molecule inhibition studies would strengthen the definitive nature of the authors' claims, specifically regarding autocrine/paracrine signaling.

3) Attempt to examine the influence of other cytoskeletal elements on perinuclear properties besides just F-actin. Even if they had no effect, this would provide some evidence for specificity of particular elements, like F-actin.

*Reviewer #1:*

Overall, this is an exciting and novel study that links nuclear mechanics to cellular differentiation. The study appears to be a collaboration among several of the main cell/nuclear mechanics groups, Mauck, Lee, Chen, etc.

Several studies have previously shown this concept empirically but this study goes a level or two deeper by making multiple measurements and examining this concept mechanistically using MSC.

The study shows that nuclei indifferentiated MSCs stiffen and become resistant to deformation. This attenuated nuclear deformation was governed by restructuring of Lamin A/C and increased heterochromatin content. This change in nuclear stiffness sensitized MSCs to mechanical loading induced calcium signaling and differentiated marker expression. This sensitization was reversed when the 'stiff' differentiated nucleus was softened, and was enhanced when the 'soft' undifferentiated nucleus was stiffened through pharmacologic treatment. These data suggest that the nucleus acts as an additional mechanism to modulate cellular mechanosensation.

*Reviewer #2:*

Summary:

The article by Heo et al. evaluates the effect of MSC differentiation and stretch on nuclear mechanics and calcium signaling. The conceptual advance results from observations that differentiation-dependent changes in chromatin condensation and nuclear structure/mechanics can alter the cellular responses to physical cues by shifting the nucleus from a strain sink to a strain concentrator. This change results in differential responses to load in the magnitude of calcium signaling. This is an interesting idea and is supported by several interesting results in the article. However, the article draws additional conclusions that are less well supported, in part because it is difficult to dissect the effects of chromatin condensation on differentiation from those on nuclear mechanics – as well as the effects on mechanosensitivity of mechanical (dynamic load) or soluble factors (TGF-β) factors. This article definitely brings forward some exciting new ideas that advance our understanding of mechanobiology, but additional experiments are needed to clarify the interpretation of the conclusions.

Significant Issues:

The most important issue is the need to more clearly discriminate nuclear mechanics from differentiation. Given that TSA affects HDAC activity, and has important effects on gene expression and differentiation, another approach is needed to discriminate the effects of nuclear softening/stiffening on cellular mechanotransduction and calcium signaling, relative to the effects on differentiation. Furthermore, Figure 4 is missing important controls that are needed to interpret these findings. In addition to verifying that the TSA and D-mannitol alter the modulus of the cells in this system using the AFM-based assay (i.e. Figure 2), data for undifferentiated cells with TSA, and differentiated cells with D-mannitol, should also be shown. The same outcomes should be assessed in each condition. It could additionally be helpful to evaluate the lamin or chromatin outcomes in these conditions.

The study is manipulating and observing events over multiple time scales – including slow processes like differentiation – and quick ones like calcium signaling. After 6 days of differentiation, TSA is administered for 24h and mannitol for 30 minutes. In some cases, load is applied in a short term manner – to assess nuclear resistance to deformation, and in others, in a long term manner. These differences are very important to interpreting the data and understanding the implications of the results. Please comment on the time scale more clearly in the Results and in the Discussion.

For the two reasons above, it is difficult to interpret the interesting data in Figure 6. The authors suggest that dynamic load is driving similar changes in nuclear architecture and cellular function as TGF-β – but more quickly (i.e. at the end of subsection “Dynamic Loading Alters MSC Nuclear Mechanics and Mechano-perception”). However, some data indicate that these are distinct processes or that they achieve similar results in lamin A or gene expression via different mechanisms. In addition to clarifying the discussion and interpretation of these results, it would be helpful to design an experiment that could more fully discriminate the extent to which these stimuli are operating in the same or in distinct pathways. In other words, can you uncouple the mechanical vs soluble mechanosensitivity or show causal epistasis more convincingly?

*Reviewer #3:*

Overall Assessment: I found this to be a compelling piece of research, with well-planned and systematic experiments. It may be a bit narrow in scope for *eLife* but if the scope fits in with the aspirations of the editors, I would recommend it strongly.

Summary:

Most studies on mechano-transduction have focused on the cell as a whole and actin cytoskeleton, while the role of nuclear architecture and how it is impacted in this process stills needs to be comprehensively elucidated. Furthermore, little is known about the manner in which soluble and physical cues are integrated to direct fate decisions, especially as the cell-state changes during the differentiation process. In this study the authors investigate how the biophysical properties of the cell and nucleus first change in response to soluble cues and microenvironment, and how these changes in the biophysical properties of the nucleus subsequently influences the mechano-transduction/sensing, to promote and preserve lineage specification.

The authors have employed a previously established biomaterial platform involving aligned nanofibrous scaffolds to guide fibrochondrogenesis and apply coordinated cell deformation. Using this platform, they have systematically illustrated that nuclear architecture/nuclear stiffness is modulated during differentiation and these changes influence subsequent mechano-sensitivity, by conducting well-planned experiments to measure nuclear deformation, peri-nuclear elastic modulus and to assess the specific role of nuclear stiffness as well as structural organization. They show that nuclear stiffening during differentiation dominates the response to subsequent external mechanical stimuli with respect to nuclear mechanics/deformation (even in presence of patent and contractile cytoskeleton). It transforms the nuclei from being a 'strain sink' to a 'stress concentrator', and increases mechano-sensitivity. Furthermore, their results indicate that nuclear stiffening is both a consequence and a mediator of differentiation, as it is manifested by soluble cues inducing differentiation, as well as mechanical stimulation by itself. While this paper provides several interesting insights, some revisions are indicated as listed below.

Suggested revisions:

1) In Figure 1, the figure caption is a bit confusing and can be clarified. In Figure 1), clarify if absolute NAR values been plotted or relative/change in NAR been plotted?

Indicate in caption that nuclear strain was applied on specific days post culture as indicated in the plot (and not continuous).

Since experiments have been performed at least three times, add error bars to Figure 1, or do these figures contain pooled data from all experiments?

2) Provide some discussion on how these findings can be applied to optimize or preserve lineage specification in biomaterial constructs.

3) Provide some discussion on possible mechano-sensors or signaling pathways (YAP etc.) that could regulate lamin levels and epigenetics to influence nuclear mechanics.

4) Elevated expression of differentiation markers has been demonstrated in response to DL. Is it possible to induce functional fibrochondrogenic differentiation solely by DL?

---

## [Author Response]

*1) The primary concern to address is the ability to discriminate differences in nuclear mechanics independent of differentiation and the large differences in time scales examined (see reviewer 2's comments below). Specifically, examination of potential differences in undifferentiated and differentiated cells both with and without TSA treatment should be performed.*

Thank you for this comment. Our investigations were carried out over several time scales. The first time scale, that of differentiation, occurs over the course of ~1 week, during which time we observed changes in nuclear structure, mechanics, and distribution of internal nucleoskeletal elements. The other time scale of interest was that which occurs in response to physical perturbation. These mechanotransductive events (i.e., calcium signaling) occur within a few seconds to minutes after load is initiated. As we show, changes in nuclear mechanics that occur over a longer term can alter how these cells interpret shorter term mechanical perturbations, altering their acute mechanotransduction response. Likewise, we show that the summation of these short time-scale mechanotransduction events will eventually change the cell phenotype, over the course of several days, as is evidenced by longer term loading resulting in the restructuring of chromatin and Lamin A/C that we observed. With respect to the impact of TSA on undifferentiated cells, and D-mannitol on differentiated cells, and in response the previous review, we have performed new studies to address these questions. Please see our responses to reviewer 2’s comments below, along with Figure 8 showing the findings of these new studies.

*2) Small molecule inhibition studies would strengthen the definitive nature of the authors' claims, specifically regarding autocrine/paracrine signaling.*

Thank you for this comment. In other parallel work, focused on the very short term (3 hours to less than 10 minutes) response to loading of naïve MSCs, we probed many of the potential mechanical signaling pathways that might be involved in this process. This work was published in Scientific Reports last year, with a follow up manuscript just published in the Biophysical Journal (Please see Heo et al., 2015, 2016). Those two studies examined chromatin condensation in response to loading, and defined some of the operative molecular pathways involved in this process. Summarizing that work, we found that purinergic signaling (i.e., ATP) is a central player in the initial events that result in chromatin condensation, and that this initial response is dependent on the cell having a sufficient level of baseline contractility so as to respond to the mechanical perturbation. Our conclusions from that set of studies was that loading applied to cells that had a minimal amount of tension in their cytoskeleton (due in part to basal activity in Smad signaling in the TGF/BMP pathway) resulted in the rapid release of ATP (from intracellular stores), which initiated calcium signaling in an autocrine and paracrine fashion (shown via conditioned media assays), ultimately changing the activity of chromatin modifying enzymes, such as EZH2, impacting the chromatin condensation state in the short term.

In the current study being considered by *eLife*, we show how the culmination of these short term loading events ‘sum up’ to restructure the nucleus in much the same way as TGF-β application to induce differentiation, and implicate the nucleus as a mediator of mechano-sensitivity in this cell type as it undergoes this longer term differentiation.

*3) Attempt to examine the influence of other cytoskeletal elements on perinuclear properties besides just F-actin. Even if they had no effect, this would provide some evidence for specificity of particular elements, like F-actin.*

Thank you for this comment. Our focus on f-actin in this study was motivated by several previous studies exploring the basic mechanisms of nuclear deformation in this cell type and in this 3D context. In a previously published work (Nathan et al., Acta Biomat, 2011), we measured nuclear deformation in naïve MSCs on aligned scaffolds in the context of blockers of f-actin, microtubules, and intermediate filaments. Only blockers of f-actin polymerization altered nuclear deformation in this context, leading to our focus in the current study. More recently, we have established that the giant isoform of Nesprin 1 (a component of the LINC complex that mediates actin connectivity to the nucleus) is responsible for nuclear deformation in this cell type (and is also downstream YAP mechano-signaling) (Driscoll et al., Biophys J, 2015). Based on these two observations, it seems that f-actin is the dominant player mediating strain transfer to the nucleus in this cell type.

*Reviewer #2:*

*[…]Significant Issues:*

*The most important issue is the need to more clearly discriminate nuclear mechanics from differentiation. Given that TSA affects HDAC activity, and has important effects on gene expression and differentiation, another approach is needed to discriminate the effects of nuclear softening/stiffening on cellular mechanotransduction and calcium signaling, relative to the effects on differentiation. Furthermore, Figure 4 is missing important controls that are needed to interpret these findings. In addition to verifying that the TSA and D-mannitol alter the modulus of the cells in this system using the AFM-based assay (i.e. Figure 2), data for undifferentiated cells with TSA, and differentiated cells with D-mannitol, should also be shown. The same outcomes should be assessed in each condition. It could additionally be helpful to evaluate the lamin or chromatin outcomes in these conditions.*

Thank you for this suggestion. We agree that TSA impacts HDAC activity, and so can have important effects on differentiation in the long term. To address this, applied the minimum dose of TSA for the shortest period of time, so as to capture how mechano-response changed in response to changes in nuclear properties. To ensure that TSA was not altering differentiation to a great extent (over this short period of application), we evaluated two canonical markers of the fibrochondrogenic lineage, aggrecan and type II collagen. Baseline expression of these two important markers in differentiated cells did not change with this short term TSA treatment (Figure 4). With respect to additional controls in Figure 4, the experiments suggested have now been performed. When TSA was applied to undifferentiated cells, only a small decrease in chromatin condensation was observed, with a correspondingly slight increase in nuclear deformability (p<0.05) in these undifferentiated cells. Conversely, when we applied D-mannitol to differentiated cells, no significant additional change in chromatin condensation was observed (p>0.05). These data suggest that, over the time scale of D-mannitol action on the nucleus, it did not condense further in these differentiated cells. Likewise, addition of D-mannitol did not further attenuate nuclear deformation. Indeed, differentiation on its own had sufficiently stiffened the nucleus such that very little deformation was occurring to begin with. This data is shown in Figure 8.

Author response image 1.(**a**) Chromatin condensation parameter (CCP) as a function of treatment group (*p<0.05 vs. Ctrl, ^†^p<0.05 vs. Ctrl/TSA, n = 24). (**b**) Normalized nuclear aspect ratio (NAR) as a function of treatment group and applied stretch [normalized to NAR values at 0% (dashed line), n = 57 ~ 65).**DOI:**
http://dx.doi.org/10.7554/eLife.18207.025

*The study is manipulating and observing events over multiple time scales – including slow processes like differentiation – and quick ones like calcium signaling. After 6 days of differentiation, TSA is administered for 24h and mannitol for 30 minutes. In some cases, load is applied in a short term manner – to assess nuclear resistance to deformation, and in others, in a long term manner. These differences are very important to interpreting the data and understanding the implications of the results. Please comment on the time scale more clearly in the Results and in the Discussion.*

Thank you for this comment. Please see our response to the editor above (general comment #1). We have carefully reviewed the manuscript to ensure that the timing of each study, and the intended outcome parameter measured are clearly defined in each section, and have specifically modified the last paragraph of the Introduction to make these time scales clear from the outset. In essence, when we were probing short term mechano-transductive events, we examined cells on the order of minutes to hours, the time course over which these pathways are generally active after mechanical perturbation. Similarly, for studies examining nuclear deformation, stretch was applied on the time scale of minutes, and the elastic deformation of the nucleus was visualized immediately at each deformation step, as an indirect measure of nuclear stiffness. When we were probing the consequence of the summation of repeated mechanical perturbations, these were applied over multiple days to provide sufficient time for the cell to respond and remodel in response the loading environment. These clarifications have been emphasized in the revised manuscript. Please see revised manuscript, Introduction, last paragraph.

*For the two reasons above, it is difficult to interpret the interesting data in Figure 6. The authors suggest that dynamic load is driving similar changes in nuclear architecture and cellular function as TGF-β – but more quickly (i.e. at the end of subsection “Dynamic Loading Alters MSC Nuclear Mechanics and Mechano-perception”). However, some data indicate that these are distinct processes or that they achieve similar results in lamin A or gene expression via different mechanisms. In addition to clarifying the discussion and interpretation of these results, it would be helpful to design an experiment that could more fully discriminate the extent to which these stimuli are operating in the same or in distinct pathways. In other words, can you uncouple the mechanical vs soluble mechanosensitivity or show causal epistasis more convincingly?*

Thank you for this excellent question. Our data in this paper and other related studies suggest that these are related, but distinct phenomenon. That is, it seems that loading and application of TGFbeta result in alterations in nuclear stiffness, and that this altered nuclear stiffness changes mechano-sensitivity in similar ways.

However, the overall state of the cell between these two conditions is by no means identical. In recent unpublished data, we carried out microarray analysis comparing the response to TGFbeta at day 4 versus the response to 2 bouts of dynamic loading (each applied for three hours per day at 3% strain at 1Hz) over 4 days. As is shown in the PCA plot in Figure 9, each loading event drives alterations in global gene expression, which accumulate and stabilizes after the second loading event. Notably, the loaded samples group in the PCA plot in a quite different location than do those samples treated with TGFbeta over this time course. Thus, it appears that while ‘load-induced’ and ‘TGFbeta induced’ nuclear reorganization have some pathways and outcomes in common, the overall state of the cell is quite different based on whether the input is mechanical or soluble factor in nature. While outside the scope of the current work, ongoing studies are focused on probing positional and methylation events across the genome using RNA-seq and chromatin immunoprecipitation to better understand these distinct cell states.

Author response image 2.PCA plot showing shifts in global gene expression in response to one loading event (DL), after a period of free swelling culture (FS) following DL, after a second loading event (2^nd^ DL), and with another free swelling period (2^nd^ FS).Note that the last two points (green and grey) reside essentially on top of one another, suggesting a permanence in expression patterns across the genome following a second loading event. These positions are quite distinct from that arrived at by TGFbeta treatment (+TGF) in the absence of load over a similar time course.**DOI:**
http://dx.doi.org/10.7554/eLife.18207.026

In addition to the above, we have also recently published data on the role of basal signaling in the TGF/BMP pathway (i.e., baseline Smad activation) as well as the potential role of released TGFbeta acting in a paracrine/autocrine fashion in the response to dynamic loading. To the first point, we found that blockers of Smad2/3 (SB431542) or Smad1/5/8 (LDN193189), which mediate TGF and BMP signaling, respectively, eliminated the early (less than 3 hour) chromatin condensation we observed in response to mechanical loading. Interestingly, these blockers also dramatically decreased cell contractility (please see Heo et al. Biophys J, 2016). Of further note, when we added functional blocking antibodies to TGFbeta during and after loading, this treatment had no impact on early nuclear remodeling. This suggested to us that basal activity (in the absence of exogenous ligand) in the canonical TGF/BMP pathways is important for establishing tension in the cytoskeleton, and so influences the ability of cells to detect mechanical perturbation.

Thus, while related in that both loading and TGF result in nuclear remodeling and stiffening, and sensitization to additional mechanical stimuli as a result, the state of the cell is quite different depending on the path traveled to achieve this nuclear remodeling.

*Reviewer #3:*

*[…]Suggested revisions:*

*1) In Figure 1, the figure caption is a bit confusing and can be clarified. In Figure 1), clarify if absolute NAR values been plotted or relative/change in NAR been plotted?*

Thank you for your careful reading of our work. We apologize not being more clear in the figure legend and have modified the text to remedy this. Also, the absolute NAR values were plotted in Figure 1, but these have been changed to a normalized NAR (subsection “Differentiation attenuates stretch-induced nuclear deformation”, first paragraph).

*Indicate in caption that nuclear strain was applied on specific days post culture as indicated in the plot (and not continuous).*

The reviewer is correct in that interpretation and we apologize for the confusion and have amended the language of the figure legend to make this clearer (subsection “Differentiation attenuates stretch-induced nuclear deformation”, first paragraph).

*Since experiments have been performed at least three times, add error bars to Figure 1, or do these figures contain pooled data from all experiments?*

The histograms shown in Figure 1 are from one representative data set. All replicate studies showed a similar attenuation in the population response, and we present this data to demonstrate simply that it is not a sub-population effect, but rather a shift in the entire population.

*2) Provide some discussion on how these findings can be applied to optimize or preserve lineage specification in biomaterial constructs.*

Thank you for this comment. We have added additional text to the Discussion to highlight this point (fifth paragraph).

3) Provide some discussion on possible mechano-sensors or signaling pathways (YAP etc.) that could regulate lamin levels and epigenetics to influence nuclear mechanics.

Again, thank you for this comment. We have added additional discussion of potential signaling pathways relevant to this nuclear remodeling phenomenon. Please see Discussion, fourth paragraph.

*4) Elevated expression of differentiation markers has been demonstrated in response to DL. Is it possible to induce functional fibrochondrogenic differentiation solely by DL?*

Please see our response to reviewer #2 above. While loading certainly drives aspects of fibrochondrogenic differentiation, including similar levels of collagen 1 and 2 gene expression, our recent microarray analysis provides evidence that these two conditions remain distinct at 4 days of loading/TGF addition. Ongoing work is probing these differences, and whether there is a convergence in the phenotype with longer time periods of loading/soluble factor addition.